# Reliability of center of pressure excursion as a measure of postural control in bipedal stance of individuals with intellectual disability: A pilot study

**Roi Charles Pineda** [1]\*, **Ralf Th Krampe** [2], **Yves Vanlandewijck** [1], **Debbie Van Biesen** [1]

**1** Department of Rehabilitation Sciences, Faculty of Movement and Rehabilitation Sciences, KU Leuven, Leuven, Belgium, **2** Brain & Cognition Group, Faculty of Psychology and Educational Sciences, KU Leuven, Leuven, Belgium

\* roicharles.pineda@kuleuven.be

## Abstract

The high prevalence of postural instability in individuals with intellectual disability (ID) warrants the need for reliable and practical postural control assessments. Stabilometry is a postural control assessment that has been widely used for clinical populations. However, the scant systematic knowledge about the reliability of stabilometric protocols for adults with ID renders results questionable and limits its value for clinicians and researchers. The study's purpose was to develop a stabilometric protocol for adults with and without ID based on optimal combinations of shortest necessary trial durations and the least number of trial repetitions that guarantee sufficient reliability. Participants performed six trials of bipedal standing in 2 vision (eyes open vs eyes closed) x 2 surface (solid vs compliant) conditions on a force platform. Several parameters were calculated from the first 10-, 20-, and 30-s interval of every center-of-pressure (COP) trial data. For different trial durations, we identified the number of trials that yielded acceptable relative (intraclass correlation coefficient $\geq 0.70$) and absolute (standard error of measurement < 20%) reliability using the Spearman-Brown prophecy formula. To determine the optimal combination of trial duration and number of repetition for each COP parameter, we implemented a two-step process: 1) identify the largest number of repetition for each of the three trial durations and then 2) select the trial duration with the lowest number of repetition. For both ID- and non-ID groups, we observed a trend whereby shorter trial durations required more repetitions and vice versa. The phase plane and ellipse area were the most and least reliable center-of-pressure parameter, respectively. To achieve acceptable reliability, four 30-s trials of each experimental condition appeared to be optimal for testing participants with and without ID alike. The results of this research can inform stabilometric test protocols of future postural control studies of adults with ID.

**Data Availability Statement:** All relevant data are within the manuscript and its Supporting Information files.

**Funding:** The work of RCP was supported by Project Grant No. G0C6817N awarded to YV by the Research Foundation - Flanders, https://www.fwo.be/en/. The funder had no role in study design, data collection and analysis, decision to publish, or preparation of the manuscript.

**Competing interests:** The authors have declared that no competing interests exist.

# Introduction

Postural control is the ability to achieve, maintain and restore one's center of mass (COM) within their base of support [1] and, as such, provides foundation for the successful and safe execution of sensorimotor tasks. This ability is the product of concerted activities of multiple body systems. Information about the body and the environment are continuously monitored by the central nervous system from visual, somatosensory and vestibular inputs. In response to postural instability, the central nervous system regulates the necessary voluntary and automatic adjustments in motor output to stabilize the body [2]. Therefore, the maintenance of postural control is contingent on intact neuromusculoskeletal functions and any disruption of the multiple systems that regulate and maintain postural control results in increased postural instability.

Intellectual disability (ID) is characterized by significant restrictions in both intellectual functioning (IQ $\leq$ 75) and adaptive behavior that have been evident before 18 years of age [3]. While intellectual impairment is the main feature of ID, motor impairments, specifically problems in postural control, are also commonly reported (e.g., [4, 5]). A systematic review by Enkelaar and colleagues [6] concluded that postural instability in individuals with ID is already evident in childhood and remains present during their entire lifespan. Furthermore, age-related decline in postural control manifests earlier in individuals with ID. As a result, adults with ID have a substantially higher incidence of falls and consequent hospitalization than the general population [7–9].

The prominence of postural instability among individuals with ID calls for its careful assessment to identify those who are at risk for falls. Moreover, reliable assessments are critical to monitor the effects of interventions designed to improve postural control. A common method for evaluating postural control is stabilometry, which is a recording of the center of pressure (COP) excursion during quiet upright stance using a force platform. COP excursion is used as a measure of postural control, as it has been demonstrated to approximate COM excursion (i.e., postural sway) under static conditions [10, 11]. However, the lack of standardization of the measurement procedure in stabilometry makes its use challenging and limits comparison between studies that have used different testing protocols [12]. This issue applies to clinical populations (including ID-individuals) as much as it does to the general population and has consequences on the reliability of results gained from stabilometry. Thus, our current study aimed to develop a reliable stabilometric protocol for evaluating postural control of adults with ID.

Ruhe, Fejer, and Walker's [12] systematic review on the intrasession reliability of static bipedal stabilometry affirmed the lack of consistency in the methods used among postural control studies. They identified several variables, namely trial duration, number of trial repetitions, experimental conditions, and chosen COP parameters, that influence reliability of stabilometric results. Longer trial durations (60 s or more) and averaging more repetitions (3–5 trials) are generally recommended for increased reliability because of the inherently wide variability of COP measurements [13, 14]. Reliability is likewise affected by the experimental conditions, including visual and surface manipulations, implemented in stabilometric studies. Compared with eyes open (EO), eyes closed (EC) conditions generally yield higher reliability values in both healthy young [15] and older adults [16]. However, EC conditions require a longer trial duration because short trials, e.g., 10 s, do not provide sufficient time for adaptation to the destabilizing effect of the loss of visual input [14, 17]. As for experimental manipulations on surface conditions, standing on a compliant surface appears to be less reliable than standing on a rigid surface [12]. Nevertheless, the inclusion of a difficult condition (e.g., compliant surface) may be desirable because challenging conditions discriminate better between age- and

pathology-related differences in postural control [18–21]. As to the difference between traditional COP parameters, measures of velocity and phase plane had the highest reliability while the ellipse area had the lowest [22, 23].

Given that reliabilities of COP measures vary depending on the participants' characteristics like age and pathology [24], recommendations derived from the aforementioned studies may not be optimal or feasible for studies on a clinical population. For instance, the recommended 60-s trial may be too long for patients with postural instability to stand still. Studies on elderly individuals [23, 25] and patients with stroke [26] and Parkinson's disease [27] reported adequate reliability of COP measures obtained from 30-s trials. These shorter trials could likewise benefit individuals with ID, as they may fatigue more easily [28] and have limited attention and motivation to complete tasks [29]. Although systematic knowledge about reliability and practicality of stabilometric protocols for adults with ID is scarce, several related studies have nonetheless been conducted (see S1 Table). These studies adopted different experimental conditions and widely varying testing specifications, ranging from 10–40 s of trial duration and 1–5 trial repetitions. There is further lack of consistency in sampling frequency of force plate data and COP-based stability parameters reported. Moreover, of these studies, only one paper published 26 years ago reported the reliability of COP measures in static bipedal stance in adults with ID. Suomi and Koceja [30] evaluated the postural control of 22 men with mild to moderate ID ($M_{age}$ = 30.3, $SD_{age}$ = 5.5 years; $M_{IQ}$ = 57.8, $SD_{IQ}$ = 9.7) and a control group of 22 men and 22 women without ID. Participants performed three 15-s trials in EO and EC conditions. The authors found that intraclass correlation coefficient (ICC) values were higher in EC ($ICC_{2,1}$ = 0.74–0.84) than in EO ($ICC_{2,1}$ = 0.52–0.67) condition. Furthermore, ICC values for both conditions were lower for men with ID compared with the two control groups (men: $ICC_{2,1}$ = 0.86–0.93; women: $ICC_{2,1}$ = 0.67–0.90). The differences in reliability may contribute to the conflicting findings on the COP characteristics of individuals with ID in the literature. For instance, Suomi and Koceja [30] showed that the postural control of adults with ID is vision dominant (i.e., greater COP excursion increase from EO to EC conditions in the ID-group than in the non-ID-group) while Blomqvist et al. [4] and Dellavia et al. [31] provided data on the contrary.

In stabilometric studies comparing two groups (i.e., group of individuals with and without ID), it is necessary that the chosen procedure to assess postural control possesses similar reliabilities for either group. Shortened trial durations may make the testing feasible for individuals with ID. Likewise, including conditions that are more challenging may be better at detecting differences between individuals with and without ID. Reliability, however, should be as important a consideration as feasibility and discriminative power when deciding on assessments and procedures of its use. Thus, a good compromise of trial duration and repetition while testing postural control in varying degrees of difficulty is essential not only to achieve acceptable reliability for both groups but also to keep testing feasible for all participants, regardless of group membership. This pilot study's primary purpose was to identify the shortest trial duration and the minimum number of repetitions that yield acceptable reliabilities of COP parameters in four (2 vision x 2 surface) conditions for adults with and without ID. We expected higher reliability to be obtained from longer trial durations and a greater number of trial repetitions. As a second goal, we aimed to determine effect sizes for the difference between the postural control of individuals with and without ID in order to inform sample size calculations of future studies.

## Materials and methods

### Participants

We recruited ten individuals into the study. Participants included five adults with a diagnosis of ID, who were identified through an assisted living facility for people with disabilities, and

five age- and sex-matched controls without ID. All ID-participants have been diagnosed with ID by a physician. The following were the inclusion criteria: (1) age 18 to 40 years; (2) no reported neurologic, orthopedic, muscular, or cardiovascular symptoms or diagnosed medical condition; (3) no history of head or lower extremity injury in the past year; (4) $\geq$ 0.5 decimal visual acuity; and (5) abstinence from alcohol at least 24 hours prior to testing. We only included young adults to avoid introducing maturation and aging effects. Additionally, even though deterioration in postural control typically arises around the age of 60 [32], we chose a lower age limit of 40 years for two reasons. First, vestibular function has been shown to diminish after the age of 40 [33]. Second, based on the modelled lifespan trajectory of postural control ability of ID-individuals by Enkelaar et al. [6], ID-individuals are likely to experience age-related postural decline earlier than non-ID-individuals.

Study design and protocol were reviewed and approved by the UZ/KU Leuven Research Ethics Committee (B322201731833/S59931). Although the ability to make informed consent may be impaired in ID-individuals, Horner-Johnson and Bailey [34] documented that ID-participants were capable of giving their own consent to participate in low risk studies and offered suggestions to ensure that they understand every step of the consent process. Following these suggestions, we read the information and consent form aloud to them and asked after each section if they understood it, if they have questions about it, and if they agree to it. We also probed for understanding by having them express in their own words what the study is about, what voluntary means, what kind of risks they will be exposed to, and what is expected of them. A staff from their respective living facility was also present throughout the testing session. All participants provided their written informed consent prior to participation in the study, in accordance with the Helsinki Declaration.

## Apparatus

We assessed postural control using a portable force platform (AccuSway, Advanced Mechanical Technology Inc. [AMTI], Watertown, MA, USA). Postural data were acquired and recorded using Balance Clinic software version 2.03.00 (AMTI) loaded on a Dell laptop. The acquisition sampling frequency was set at 100 Hz and was filtered using a fourth-order zero phase Butterworth low-pass filter with a cut-off frequency of 10 Hz [11, 12]. Four experimental conditions, with two vision by two surface conditions were included. Participants either stood directly on the force platform (solid surface) or on a 50 x 41 x 6 cm foam (Airex® Balance Pad, Airex AG, Sins, Switzerland) placed on top of the force platform (compliant surface). The foam pad has a density of 55 kg/cm$^3$ and tensile strength of 260 kPa [35]. We tested the participants on both surface conditions with blindfolds (EC) or without (EO).

## Experimental protocol

Before enrolling prospective participants into the study, we performed a phone screening where we asked participants and/or the staff nurse of the facility (in the case of ID-individuals) about demographic information, recent injuries and current health status, and regular physical activities. On the day of testing, we measured the participants' height, weight, and foot length, and tested their visual acuity using a tumbling E chart. We also confirmed from self-report whether participants had consumed alcohol within 24 hours prior the test.

Each participant was tested in a single session, which lasted about 2 hours. We collected force plate data from two blocks of testing. Each block consisted of three consecutive 35-s trials in each condition, yielding 24 trials (3 trials x 4 conditions x 2 blocks). In the first block, the sequence was as follows: EO solid surface (EO-S), EC solid surface (EC-S), EO foam surface (EO-F), and EC foam surface (EC-F). This sequence was reversed in the second testing block.

A 35-s trial duration was adopted after considering the available evidence that 30-s trials may be adequate for a clinical population, as well as to limit total testing duration. Test order was fixed and was identical for all participants. Participants sat for 1 minute on a chair to rest between trials. Between the two postural control blocks, we administered the performance sub-scale of the Wechsler Abbreviated Scale of Intelligence (WAIS) [36]. We tested the participants' IQ to provide a measure to contrast intellectual function between the two groups and because we had no access to the ID-participants' IQ scores (due to the facility's privacy restrictions). Prior to the actual postural tests, we familiarized participants with the tasks during two 30-s warm-up trials for each condition.

During testing, participants stood barefoot at hip-width apart and with big toes pointing forward. After establishing proper foot positioning, we drew an outline of both feet to keep foot placement consistent across trials on both solid and compliant surfaces. Furthermore, intermalleolar distance was checked and maintained throughout all the trials. Before each trial, we instructed the participants to stand as still as possible with their arms to their sides. They were also instructed to look straight ahead at a 3-cm red circle located on the wall at eye level.

## Data analysis

The initial and final 2.5 s of each of the 35-s COP time series were considered padding points to minimize amplitude distortion from data filtering and were excluded from further analysis [37]. We used four traditional COP parameters: mean COP velocity and displacement (antero-posterior [AP] and mediolateral [ML] directions), 95% confidence ellipse area, and phase plane. Phase plane is a stability parameter that incorporates the position and velocity of COP and was computed based on the combined stability parameter described by Riley, Benda, Gill-Body, and Krebs [38]. The three other COP excursion parameters, amplitude (average displacement from the mean COP), velocity (total length of the COP path per unit time), and ellipse area (area of the ellipse that captures 95% of the data points), were based on formulae used by Prieto, Myklebust, Hoffman, Lovett, and Myklebust [39]. We calculated these COP parameters from the remaining 30 s COP time series data, as well as from the first 10 s and 20 s of the trial in order to explore whether it is possible to shorten trial duration further without sacrificing reliability. To determine within-session learning and fatigue effects, we compared the COP parameters from the first three trials in each condition with the last three trials using a paired $t$-test. No statistically significant difference emerged ($\alpha$-level at 95%). Thus, we decided to pool all six trials for subsequent analyses.

Reliability estimates of COP parameters obtained at the level of single trials were computed using $ICC_{2,1}$, a 2-way random effects model with an agreement coefficient [40]. This model of ICC compares within-subject variability with between-subject variability, considering random effects over time. Five participants and six trials provide 80% power to differentiate ICC values between 0.70 and 0.95 with type 1 error set at 0.05 [41]. Given the pilot nature of the study, we did not apply $\alpha$-value correction for multiple testing. Because the ICC only measures relative reliability, we also calculated the standard error of the measurement (SEM) as an estimate of absolute reliability [42]. We computed SEM, expressed in percentage, by dividing the product of the standard deviation (SD) and square root of (1 –$ICC$) by the mean and multiplying it by 100.

As a next step, we implemented the statistical method described by Lafond and colleagues [22]. For each trial duration (i.e., 10, 20, and 30 s), we used the Spearman-Brown prophecy formula (1) to identify the number of trials ($k$) that, when averaged, yielded an acceptable relative (ICC $\geq$ 0.70 [43]) and absolute (SEM $\leq$ 20 [23]) reliability estimate. We also computed the reliability coefficient of the average of increasing $k$ repeated trials ($R_k$) with a single-measure

reliability coefficient $R$, i.e., $ICC_{2,1}$ using the same formula.

$$R_k = \frac{kR}{1 + (k-1)R}$$ (1)

To determine the optimal trial duration and repetition for each COP parameter, we implemented a two-step process. First, for each of the three trial durations, we identified the largest $k$ value from all four postural control conditions of both groups. Then, we selected the trial duration with the lowest $k$ value. We chose to do it in this manner because, ideally, the chosen COP parameter should possess acceptable reliability for all the experimental conditions and for both groups.

Lastly, we performed independent $t$-tests between the two groups, using the COP parameters with acceptable reliability as the dependent variable. From the $t$-statistic and the sample sizes of the ID- ($n_1$) and non-ID- ($n_2$) group, we calculated Cohen's $d$:

$$\text{Cohen's } d = t\sqrt{\frac{n_1 + n_2}{n_1 n_2}}$$ (2)

Due to the study's small sample size, we used Hedge's $g$ as the corrected effect size [44].

$$\text{Hedge's } g = \text{Cohen's } d \left(1 - \frac{3}{4(n_1 + n_2) - 9}\right)$$ (3)

Using the calculated effect size, a post hoc sample size calculation was performed in G*Power version 3.1.9.4 for Windows [45]. All other statistical analyses were performed with Statistical Package for the Social Sciences (SPSS), PC Statistical Package version 25.0 for Windows (SPSS, Inc., Chicago, IL, USA).

## Results

Description of the participants' characteristics are summarized in Table 1. None of the ID-participants had Down syndrome. All participants were able to complete the testing protocol. During testing, we documented practical testing issues that are worth reporting. We found it challenging to standardize the foot position of the participants with ID when we resumed testing after every rest break. They had problems placing their feet within the foot tracings or kept moving their feet after it had already been correctly positioned. Difficulties standardizing foot placement were even greater for the standing trials on foam. During the 35-s duration of the actual standing trials, however, we did not encounter any problems. Considering these problems and the fact that all participants could easily remain standing for 35 s, fewer trial

**Table 1. Participant characteristics.**

| Characteristics¤ | ID group ($n = 5$)¤ | Non-ID group ($n = 5$)¤ |
|---|---|---|
| $M$ Age (years)¤ | 29.20 (7.92)¤ | 29.00 (7.62)¤ |
| Sex (female / male)¤ | 1 / 4¤ | 1 / 4¤ |
| $M$ Foot length (cm)¤ | 26.50 (1.38)¤ | 26.44 (1.40)¤ |
| $M$ Performance IQ¤ | 75.80 (10.0)¤ | 115.60 (7.9)¤ |
| $M$ BMI (kg/m$^2$)¤ | 25.98 (4.3)¤ | 22.25 (2.1)¤ |
| $M$ Weekly PA (hours)¤ | 2.90 (2.1)¤ | 3.05 (1.1)¤ |

Figures enclosed in parentheses are standard deviations; ID, intellectual disability; IQ, intelligence quotient; BMI, body mass index; PA, physical activity

**Table 2. Number of trials (k) that would yield an $ICC_{2,k} \geq 0.70$ and SEM% $\leq 20$ and corresponding $ICC_{2,k}$ and SEM% for trial lengths of 10, 20, and 30 s.**

| Center of pressure parameters | Participants with intellectual disability | | | | | | | | | Participants without intellectual disability | | | | | | | | |
|---|---|---|---|---|---|---|---|---|---|---|---|---|---|---|---|---|---|---|
| | 10-s trial | | | 20-s trial | | | 30-s trial | | | 10-s trial | | | 20-s trial | | | 30-s trial | | |
| | k | ICC | SEM | k | ICC | SEM | k | ICC | SEM | k | ICC | SEM | k | ICC | SEM | k | ICC | SEM |
| *M* amplitude ML | | | | | | | | | | | | | | | | | | |
| →EO-S | 3 | 0.83 | 19.9 | 3 | 0.86 | 18.7 | 3 | 0.91 | 19.2 | 9 | 0.86 | 19.5 | 4 | 0.83 | 19.4 | 6 | 0.82 | 19.4 |
| →EC-S | 13 | 0.90 | 19.4 | 12 | 0.90 | 19.9 | 6 | 0.90 | 18.8 | 3 | 0.77 | 19.7 | 2 | 0.81 | 16.2 | 3 | 0.82 | 17.3 |
| →EO-F | 6 | 0.71 | 19.4 | 4 | 0.75 | 16.4 | 3 | 0.75 | 15.2 | 15 | 0.81 | 19.5 | 4 | 0.74 | 16.8 | 3 | 0.76 | 17.7 |
| →EC-F | 4 | 0.73 | 14.7 | 3 | 0.75 | 12.7 | 4 | 0.75 | 15.3 | 1 | 0.76 | 17.6 | 1 | 0.77 | 16.1 | 1 | 0.80 | 14.8 |
| *M* amplitude AP | | | | | | | | | | | | | | | | | | |
| →EO-S | 14 | 0.71 | 16.6 | 5 | 0.72 | 14.9 | 3 | 0.70 | 16.2 | 16 | 0.86 | 19.5 | 5 | 0.83 | 18.9 | 4 | 0.80 | 19.0 |
| →EC-S* | — | — | — | 15 | 0.70 | 16.5 | 12 | 0.72 | 14.4 | — | — | — | — | — | — | — | — | — |
| →EO-F | 2 | 0.75 | 16.2 | 2 | 0.75 | 16.2 | 1 | 0.76 | 14.9 | 20 | 0.84 | 19.9 | 9 | 0.72 | 19.2 | 5 | 0.74 | 19.7 |
| →EC-F | 5 | 0.71 | 17.3 | 2 | 0.74 | 16.2 | 2 | 0.78 | 14.7 | 3 | 0.73 | 18.0 | 2 | 0.80 | 13.7 | 3 | 0.75 | 14.7 |
| *M* velocity ML | | | | | | | | | | | | | | | | | | |
| →EO-S | 2 | 0.80 | 16.1 | 2 | 0.79 | 15.2 | 2 | 0.80 | 12.9 | 2 | 0.72 | 16.9 | 2 | 0.82 | 13.0 | 1 | 0.72 | 14.1 |
| →EC-S | 28 | 0.89 | 19.8 | 7 | 0.85 | 19.7 | 3 | 0.83 | 18.5 | 3 | 0.77 | 14.4 | 2 | 0.70 | 13.9 | 2 | 0.77 | 11.7 |
| →EO-F | 32 | 0.70 | 18.4 | 4 | 0.71 | 12.2 | 3 | 0.72 | 9.8 | 4 | 0.74 | 16.9 | 3 | 0.76 | 16.5 | 2 | 0.72 | 16.5 |
| →EC-F | 7 | 0.73 | 13.7 | 2 | 0.73 | 10.3 | 4 | 0.71 | 13.1 | 2 | 0.71 | 19.6 | 2 | 0.79 | 16.3 | 1 | 0.78 | 15.6 |
| *M* velocity AP | | | | | | | | | | | | | | | | | | |
| →EO-S | 2 | 0.82 | 11.6 | 2 | 0.82 | 10.9 | 1 | 0.79 | 12.2 | 1 | 0.83 | 14.4 | 1 | 0.80 | 14.6 | 1 | 0.82 | 13.2 |
| →EC-S | 7 | 0.81 | 19.3 | 3 | 0.79 | 18.1 | 2 | 0.84 | 15.9 | 6 | 0.73 | 17.1 | 2 | 0.78 | 14.7 | 1 | 0.73 | 14.9 |
| →EO-F | 2 | 0.73 | 13.2 | 3 | 0.76 | 12.0 | 2 | 0.71 | 13.4 | 3 | 0.77 | 16.1 | 1 | 0.70 | 17.4 | 1 | 0.74 | 15.0 |
| →EC-F | 3 | 0.76 | 11.3 | 1 | 0.74 | 10.3 | 1 | 0.78 | 9.7 | 2 | 0.75 | 16.0 | 2 | 0.79 | 14.7 | 1 | 0.75 | 15.1 |
| Ellipse area | | | | | | | | | | | | | | | | | | |
| →EO-S | 8 | 0.89 | 19.0 | 10 | 0.94 | 19.2 | 5 | 0.93 | 19.8 | 25 | 0.94 | 19.8 | 11 | 0.94 | 19.1 | 12 | 0.93 | 19.4 |
| →EC-S | 216 | 0.98 | 19.9 | 59 | 0.97 | 19.8 | 15 | 0.95 | 19.7 | 219 | 0.89 | 19.9 | 3 | 0.83 | 18.7 | 5 | 0.82 | 19.9 |
| →EO-F | 10 | 0.94 | 19.1 | 3 | 0.90 | 18.2 | 3 | 0.89 | 18.7 | 35 | 0.94 | 19.9 | 4 | 0.82 | 18.2 | 2 | 0.91 | 15.0 |
| →EC-F | 38 | 0.88 | 19.8 | 3 | 0.80 | 19.1 | 12 | 0.89 | 19.7 | 5 | 0.88 | 18.5 | 3 | 0.88 | 17.2 | 2 | 0.85 | 18.6 |
| Phase plane | | | | | | | | | | | | | | | | | | |
| →EO-S | 3 | 0.75 | 15.6 | 2 | 0.76 | 13.3 | 1 | 0.75 | 14.2 | 1 | 0.79 | 12.5 | 1 | 0.81 | 11.9 | 1 | 0.79 | 11.4 |
| →EC-S | 9 | 0.86 | 19.2 | 4 | 0.80 | 18.9 | 2 | 0.79 | 18.4 | 4 | 0.72 | 14.1 | 2 | 0.75 | 11.9 | 2 | 0.73 | 11.0 |
| →EO-F | 3 | 0.75 | 15.1 | 2 | 0.77 | 11.8 | 2 | 0.78 | 12.0 | 4 | 0.71 | 16.3 | 3 | 0.77 | 12.7 | 2 | 0.74 | 12.7 |
| →EC-F | 4 | 0.73 | 11.8 | 2 | 0.72 | 10.8 | 2 | 0.81 | 11.1 | 3 | 0.79 | 14.9 | 2 | 0.77 | 13.0 | 1 | 0.73 | 12.4 |

*$k$, $ICC_{2,k}$ and SEM% cannot be calculated when $ICC_{2,1} \leq 0$; ML, mediolateral; AP, anteroposterior; EO-S, eyes open, solid surface; EC-S, eyes closed, solid surface; EO-F, eyes open, foam surface; EC-F, eyes closed, foam surface

repetitions of longer trial durations appeared to be more convenient than more repetitions of shorter trial durations.

The *k* repetitions necessary to obtain an ICC $\geq 0.70$ and SEM $\leq 20$ for every trial duration, as well as the corresponding $ICC_{2,k}$ and SEM are listed in Table 2. In general, we observed a trend of decreasing *k* with longer trial durations. Thus, shorter trial durations had to be performed with more repetitions while longer trial durations required fewer repetitions to achieve the same level of reliability. This is true for both groups.

Ellipse area was the least reliable COP parameter as, even at the longest trial duration of 30 s, 15 repetitions were needed to meet our threshold for acceptable reliability. In contrast, the most consistently reliable COP parameter was the phase plane. Good reliability estimates were obtained with two 30-s trials. For mean ML and AP velocity, the most optimal combination

**Table 3. Means and standard deviations of center of pressure parameters of participants with and without intellectual disability (ID), as well as values for *t*-statistic, significance (*p*), and effect size (Hedge's *g*).**

| | with ID | | without ID | | | | |
|---|---|---|---|---|---|---|---|
| | *M* | *SD* | *M* | *SD* | *t* | *p* | Hedge's *g* |
| *M* velocity ML (mm/s) | | | | | | | |
| →EO-S | 4.6 | 1.2 | 4.8 | 1.2 | -0.29 | 0.78 | 0.17 |
| →EC-S | 5.5 | 2.1 | 5.0 | 1.1 | 0.40 | 0.70 | 0.23 |
| →EO-F | 9.9 | 1.4 | 9.5 | 2.4 | 0.30 | 0.77 | 0.16 |
| →EC-F | 16.1 | 2.8 | 15.3 | 4.9 | 0.30 | 0.77 | 0.17 |
| *M* velocity AP (mm/s) | | | | | | | |
| →EO-S | 8.6 | 2.3 | 7.3 | 2.3 | 0.96 | 0.37 | 0.55 |
| →EC-S | 11.7 | 4.3 | 8.8 | 2.3 | 1.3 | 0.22 | 0.76 |
| →EO-F | 16.8 | 3.4 | 14.8 | 4.1 | 0.85 | 0.42 | 0.49 |
| →EC-F | 27.7 | 5.5 | 22.4 | 6.5 | 1.4 | 0.21 | 0.78 |
| Phase plane (unitless) | | | | | | | |
| →EO-S | 10.3 | 2.8 | 8.5 | 2.0 | 1.1 | 0.30 | 0.64 |
| →EC-S | 13.4 | 4.8 | 9.8 | 1.8 | 1.6 | 0.15 | 0.90 |
| →EO-F | 20.8 | 4.7 | 18.9 | 4.0 | 0.69 | 0.51 | 0.40 |
| →EC-F | 32.8 | 7.6 | 26.9 | 6.1 | 1.4 | 0.21 | 0.77 |

ML = mediolateral; AP = anteroposterior; EO-S = eyes open, solid surface; EC-S = eyes closed, solid surface; EO-F = eyes open, foam surface; EC-F = eyes closed, foam surface

was four 30-s trials, while six 30-s trials were necessary for mean amplitude in the ML and AP direction, excluding the mean AP amplitude in the EC-S condition. A value for *k* could not be calculated for mean AP amplitude in the EC-S condition because the $ICC_{2,1} < 0$ in all the trial durations for the non-ID-group. This can be indicative of poor reliability.

Table 3 compares the ID- and non-ID-group on COP parameters having acceptable reliability. Although we observed no significant differences ($p < 0.05$), phase plane in the EC-S condition demonstrated large effect sizes (Hedge's $g > 0.80$) while phase plane in EC-F condition and mean AP velocity in the EC-S and EC-F conditions have effect sizes approaching .80. Using the most optimistic effect size of 0.90, a minimum sample size of 32 participants (16 per group) should have 80% power to detect differences in postural control between ID- and non-ID-group with α = 0.05.

## Discussion

The purpose of this pilot study was to identify the optimal combination of trial durations and number of trials to meet threshold criteria for COP-parameter reliabilities of ICC ≥ 0.70 and SEM ≤ 20. For the two most reliable COP parameters (i.e., mean velocity and phase plane), four 30-s trials turned out to be the optimal combination to meet our criteria. This combination applies to both ID- and non-ID-group and across all postural control conditions. Shorter trial durations (e.g., 10 and 20 s) were less optimal. Any testing time savings gained from shorter trial durations would be more than outweighed by the number of trial repetitions necessary to achieve similar reliabilities as with our optimal combination.

From our results, each COP parameter required different numbers of repetitions reflecting the parameters' reliabilities. For mean COP amplitude in the AP and ML directions to be reliable in all conditions, 30-s trials had to be repeated six times. These findings are at odds with the lone reliability study on COP measures in men with ID by Suomi and Koceja [30]. In their

study, three 15-s trials were sufficient for the COP amplitude in the AP and ML directions to obtain the same reliability criteria that we set for this pilot study (note that only EO-S and EC-S conditions were tested in their study). Several explanations could account for this incongruent finding. Suomi and Koceja reported standard deviations of the COP amplitude, as opposed to mean COP amplitude that we described in our study. However, it is unlikely the sole reason for the discrepancy because existing studies (e.g., [23, 46]) have calculated very similar reliability coefficients between the mean and standard deviation of COP amplitude. Another possible explanation is related to the difference in sampling and cut-off frequency of the raw COP data. This pilot study used a sampling rate of 100 Hz with a 10 Hz cut-off while Suomi and Koceja's study used a 50 Hz sampling frequency and a cut-off frequency of 5 Hz. In Ruhe and colleagues' [12] systematic review on the reliability of stabilometry, they determined that cut-off and, to a lesser extent, sampling frequency significantly affects the reliability of COP data. Further, they recommended a sampling frequency of 100 Hz with a 10 Hz cut-off for traditional COP parameters like amplitude, velocity and ellipse area. We also found that mean AP amplitude in the EC-S condition had negative ICC values. While this suggests that this specific COP measure is not reliable, existing literature (e.g., [12]) does not support this. An alternative interpretation would be to consider this negative ICC as a spurious result arising from the low sample size of our study. We entertain this possibility considering that it significantly differed from the ICC values calculated in the other postural control conditions. Furthermore, previous studies on young adults with and without ID have reported higher reliability coefficients for AP amplitude in EC-S conditions [15, 30, 47].

Among the COP parameters we evaluated, ellipse area had the worst reliability for both the ID- and non-ID-group. Previous studies on healthy young and older adults [48] and young adults with musculoskeletal disorders [23] also found poor ICC values for ellipse area. The three 30-s trial protocol implemented in these studies was insufficient to yield ICCs $\geq$ 0.70 in EO-S and EC-S conditions. The inadequacy of three repetitions to achieve acceptable reliability on both EO-F and EC-F was also reported in a study on young gymnasts even when the trial duration was 120 s [49]. Our pilot study's findings taken together with existing literature provide evidence against the use of ellipse area as a measure of postural control, especially for short trial durations.

In contrast, mean COP velocity and phase plane possessed the highest reliability, which agrees with multiple studies on clinical and non-clinical population without ID [12, 23, 25, 50]. Furthermore, we found that four 30-s trials yield reliable data on COP velocity and phase plane. Two studies involving healthy young adults produced conflicting results on the reliability of mean COP velocity in EC-S conditions. For two 30-s trials, Takala and colleagues [47] reported a reliability coefficient below our threshold (ICC = 0.46–0.54) while Pinsault and Vuillerme [51] obtained ICCs > 0.70. Our findings are in agreement with Pinsault and Vuillerme's results, as we also found that two 30-s trials were adequate to obtain reliable data on mean COP velocity for our non-ID group. Note that we used the same form of reliability coefficient (i.e., $ICC_{2,1}$) as did Pinsault and Vuillerme in their study. Takala et al. [47] did not explicate their version of ICC and it is possible that related differences can account for discrepancies in the findings. In our study, four trials were necessary for the ID-group to obtain the same level of reliability as the non-ID-group. We found no studies reporting reliability of mean COP velocity obtained from four 30-s trials in participants with ID. The greater number of repetitions needed for the ID-group may be related to the higher performance variability typically observed within this population [52, 53]. The problem of large performance variability of COP parameters causing lower ICC values in non-ID clinical populations has been documented by Harringe and colleagues [49].

For phase plane, two repetitions of 30-s trial were enough to reach acceptable reliabilities for the ID- and non-ID-group. Raymakers, Samson, and Verhaar [54] also demonstrated that

two trials were adequate for reliable phase plane data in adults without ID. However, they used 50-s trials in the EO-S condition only and used the coefficient of variation as a reliability coefficient, which limits comparability between their study and ours. A study on young adults with anterior cruciate ligament injury by Hadian et al. [50] reported ICC values of the phase plane parameter in the EO-S, EC-S, and EC-F conditions. They showed that three 30-s trials were required for ICC $\geq$ 0.70 and SEM $\leq$ 20. This is one repetition more than what our findings suggested but the difference in study population may explain the dissimilarity.

Based on our findings, reliable COP measures could be obtained with trial durations even shorter than 30 s albeit at the expense of increased numbers of trial repetition. However, increasing the number of repetitions to compensate for shorter trials has its own shortcomings. First, the increased number of repetitions for shorter trials may lead to a longer active testing time (i.e., time when the force plate is recording the COP excursion) than longer trials and fewer repetitions. For instance, our results showed that mean COP velocity needed four, seven and 32 repetitions of 30-, 20- and 10-s trial durations, respectively. Computing for active testing time (4 x 30 = 120 s; 7 x 20 = 140 s; 32 x 10 = 320 s), we found that we could save the most time with the longest trial duration. This applies, as long as the longest trial duration is feasible for the participants. Second, more between-trial periods also added to passive testing time (i.e., time before and after trials). For our participants with ID, the passive testing time was often longer because standardizing their foot position and reminding them of the instruction (e.g., stand still, keep quiet, and maintain forward gaze) needed multiple repetitions. Lastly, shorter trial durations do not provide adequate time for the postural system to adapt to the postural control challenge, especially for more difficult standing conditions [17], and it is more likely to miss lower frequency component of the COP signal [14]. Therefore, a trial duration of 30 s appears adequate to yield reliable data of certain COP parameters. This is supported by studies on healthy young adults [17, 23], as well as on adults with ID [30] and other clinical population [25–27].

In sum, the lack of consistency in stabilometric test protocols and the lack of systematic reliability studies targeting assessments of postural control in individuals with ID may explain conflicting results in the literature. To our knowledge, our study is the first to use an appropriate statistical approach [22] to determine the optimal configuration of trial duration and repetition that yields acceptable relative and absolute intrasession reliability in adults with ID. It was found that four 30-s trials are adequate to achieve acceptable reliability of COP parameters, particularly mean COP velocity and phase plane. We have performed this analysis simultaneously on both clinical and non-clinical population (i.e. ID- and non-ID-group) to ensure that the trial duration and repetition combination provides reliable COP parameters in both groups. As to our second goal, the effect sizes from our current study informed us than no fewer than 32 participants (16 participants x 2 groups) are needed to have sufficient statistical power for the detection of postural control differences between adults with and without ID.

Our results provide an important first-step in the development of a stabilometric testing protocols that are reliable and feasible for future studies. As with the general population, establishing a standard procedure for stabilometry in the ID population could strengthen its potential to not only identify individuals with postural instability but also evaluate accurately the effectiveness of postural control interventions. Furthermore, unreliable outcome measures are unlikely to discriminate those with and without postural control issues and are likely to give incorrect information on effectiveness of intervention. Only through the standardization of stabilometry and identifying reliable COP parameters could we establish stabilometric reference data that are comparable between studies. Ultimately, this would help to complete our understanding of the postural control problems experienced by individuals with ID.

It is important to acknowledge that this pilot study's findings must be interpreted with caution due to several limitations. The most important one is the small sample size, as well as representativeness of our sample. Our ID-participants were generally young, healthy, active, and high functioning individuals. Thus, applicability of our findings to older adults with more severe ID and comorbid health problems may be limited. In particular, individuals with more severe ID, who often have worse cognitive and motor skills and are more likely to have comorbid health conditions [3, 55], could have limited ability to complete stabilometric tests even for shorter durations and with fewer repetitions. Feasibility of stabilometry may be a greater concern than reliability in severe impairments [56]. The heterogeneity of sex within the groups may have introduced the increased variance in the COP measures. Lastly, we also limited our analysis to traditional linear COP parameters and did not include nonlinear parameters (e.g., sample entropy, fractal dimension, etc.). Keeping these shortcomings in mind, we consider the reported data preliminary and recommend a confirmatory reliability study with a larger sample size. Nevertheless, the findings of this pilot study could guide decisions on testing protocols for stabilometric studies among individuals with mild ID. This is particularly helpful given the limited applicability of the recommended optimal trial duration (i.e., 60 s) for reliable COP parameters used in stabilometry for adults with ID.

## Supporting information

**S1 Table. Summary of stabilometric studies in static bipedal stance on a sample that includes adolescents and adults with intellectual disability (ID).**
(DOCX)

**S1 Dataset. Trial data of the calculated COP parameters per trial duration, trial repetition number, and experimental condition.**
(XLSX)

## Acknowledgments

We would like to thank all the individuals and their residential managers for participating in our study. We also thank Jos Vanrenterghem for his valuable guidance on processing and calculation of COP data, and Julie Latin and Sofie Haesen for their help in participant recruitment and data collection.

## Author Contributions

**Conceptualization:** Roi Charles Pineda, Ralf Th Krampe, Yves Vanlandewijck, Debbie Van Biesen.

**Data curation:** Roi Charles Pineda.

**Formal analysis:** Roi Charles Pineda, Ralf Th Krampe.

**Funding acquisition:** Yves Vanlandewijck, Debbie Van Biesen.

**Investigation:** Roi Charles Pineda.

**Methodology:** Roi Charles Pineda, Ralf Th Krampe, Yves Vanlandewijck, Debbie Van Biesen.

**Resources:** Ralf Th Krampe, Yves Vanlandewijck.

**Supervision:** Debbie Van Biesen.

**Writing – original draft:** Roi Charles Pineda.

**Writing – review & editing:** Roi Charles Pineda, Ralf Th Krampe, Yves Vanlandewijck, Debbie Van Biesen.

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
