## [Decision Letter · Decision Letter 0]

23 Jul 2020

PONE-D-20-19187

Reliability of center of pressure excursion as a measure of postural control in bipedal stance of individuals with intellectual disability: A pilot study

PLOS ONE

Dear Dr. Pineda,

Thank you for submitting your manuscript to PLOS ONE. After careful consideration, we feel that it has merit but does not fully meet PLOS ONE’s publication criteria as it currently stands. Therefore, we invite you to submit a revised version of the manuscript that addresses the points raised during the review process.

The comments from the two reviewers are straightforward and may be easily addressed. Please take a close look to the comments and address each comment in a response letter. 

We look forward to receiving your revised manuscript.

Kind regards,

Robert Didden

Academic Editor

PLOS ONE

2. Please describe in your methods section how capacity to consent was determined for the participants in this study.

Reviewers' comments:

Reviewer's Responses to Questions

**Comments to the Author**

1. Is the manuscript technically sound, and do the data support the conclusions?

Reviewer #1: Yes

Reviewer #2: Yes

2. Has the statistical analysis been performed appropriately and rigorously? 

Reviewer #1: Yes

Reviewer #2: Yes

3. Have the authors made all data underlying the findings in their manuscript fully available?

Reviewer #1: Yes

Reviewer #2: Yes

4. Is the manuscript presented in an intelligible fashion and written in standard English?

Reviewer #1: Yes

Reviewer #2: Yes

5. Review Comments to the Author

Reviewer #1: The authors of the article entitled “Reliability of center of pressure excursion as a measure of postural control in bipedal stance of individuals with intellectual disability: A pilot study” aimed to test reliability of different stabilometric protocols for individuals with intellectual disability (ID) based on optimal combinations of shortest necessary trial durations and the least number of trial repetitions that guarantee sufficient reliability. The authors found that to achieve acceptable reliability, four 30-s trials of each experimental condition appeared to be optimal for testing participants both individuals with and without ID.

The study topic is interesting and may contribute to improve postural control investigation accuracy in individuals with ID. In general, the study is well designed. However, the authors have to respond to some comments regarding the manuscript. Also, there are some sentences that require rewriting and some statements that need to be addressed and clarified.

Abstract:

Authors are encouraged to more describe the method in the abstract. If there was a word limit condition you can shorten the Background of the Abstract.

Introduction

I think that the use of tables to summarize literature is not appropriate in original article.

Line 64: Because this also has consequences on the reliability of results gained from stabilometry, our current study aimed to develop a reliable stabilometric protocol for evaluating postural control of adults with ID. This sentence needs to be reworded.

Line 73: This applies to individuals with ID who may fatigue more easily [15] and have limited attention and motivation to complete tasks [16]. This sentence needs to be reworded.

Line 92: please replace with by including

Method

Postural control may be influenced by sex. So, including both male and female could be the origin of high standard deviation.

Postural control may be influenced also by the foot size. Please provide information.

Furthermore, the groups were not BMI matched that could influence results.

For individuals with ID, authors must obtain written informed consent from parents or legal garden.

Having history of injury in the past 12 month requiring medical attention must be one of exclusion criteria in postural control exploration.

Line 144: Please replace was by were

Line 147: We had four experimental conditions, with two vision by two surface conditions. Please rephrase this sentence; use the passive form

Line 151-153: This detail must be placed in the “Participant” section. Furthermore, if I understood correctly, you realized the IQ assessment using the Wechsler Abbreviated Scale of Intelligence. So, you must present this in the experimental design as a part of your study.

Please specify what participants do during the one minute of rest (standing; seated).

Please specify if all participants have the same foot positioning on the platform and the foam surface.

Please specify the characteristics of foam surface

Results

Line 236: Please consider “Table 4 compares the ID- and the non-ID-group on COP parameters having acceptable reliability.

Tables

All tables must be at the same form with the table 3

Reviewer #2: Compliments to the authors for paying attention to this important topic. Balance issues are a large problem in people with ID, as well as suitable measuring instruments. This study into measures of postural control is of relevance for future studies into this topic.

Points below have to be addressed to allow the reader to better understand the choices made in the study, the procedures and some details.

Introduction

Overall, the introduction is quite long, and I think it is possible to reduce the length a bit and make it more to the point. For example, the parts about the parameters that influence reliability are informative, but because of the length deviate the attention from the relevance of this study for people with ID. I think making this part a bit more to the point will help in strengthening the introduction.

Some additional points for the introduction:

- Page 3, line 62-64: It is not clear from the text that the lack of standardization in measurement procedures is an issue in the general population or specifically in people with ID. By reading the rest of the introduction I assume both, but it is important to specify this, throughout the text.

- Page 3, line 73: based on what findings is a trial length of 30s specified for clinical populations?

- Table 1 can be omitted from the introduction. If this was a review providing a complete overview of the studies this would be relevant, however this overview is not complete as stated by the authors. It is sufficient to describe the differences seen in test protocols in people with ID (Sampling frequency, trial length, experimental conditions, outcome parameters) in the text.

Methods

- Page 8, line 129: Why was age 18-30 years a inclusion criteria? Why this focus on relatively young adults?

- Page 9, table 2: Table 2 should be presented and described in the result section. The IQ scores of the ID group are quite high, especially because an IQ of 70 (max 75) is indicative of a significant limitation in intellectual functioning. Does this group really represent a group with ID? How was the ID classified, and people with ID selected?

- Page 9, line 139: please add the reference number of the ethical committee to this protocol.

- Page 9, line 152-153: did you did an assessment with performance subscale of the Wechsler Scale, or was this score reported from for example the participant files?

- Page 9, apparatus and tests: Also report how the other data that was collected; age, sex, BMI, weekly PA hours. All collected data and outcome measures must be described. It will also be useful to describe the outcome measures of the stabilometry with an explanation of them. An average reader, who is not fully aware of these measurements will need an explanation of these outcome measures.

- Why were these specifications chosen for the measurement protocol (35s trials, 100 Hz sampling frequency)? From the introduction we have learned that a lot of different specifications are used, so please report why these were specifically chosen? Especially, why was the 35s trial chosen, since 60s trails were found to be more reliable? Because one of the aims is to assess which duration would be most suitable and reliable, I would expect that longer trials would have been included as well.

- Page 11, line 220: specify for what comparisons t-test were performed.

Results

- The total test duration was quite long, 2hrs. Did people with ID need more rest than the 1 min between trials?

Discussion

- How representative are these results for the group of people with ID. It is mentioned that results may not be applicable to people with more severe ID, however they may also not be applicable to older adults for the same reasons (worse motor control and skills, more cognitive problems), and perhaps those with moderate levels of ID, behavioral problems etc. The study sample seems to be quite a high functioning group of people with ID (IQ around 75 and quite active) which limits the representativeness to other groups of people with ID, especially since this sample only included 5 participants with ID. This limitation should be made more clear in the discussion.

6. PLOS authors have the option to publish the peer review history of their article (what does this mean?). If published, this will include your full peer review and any attached files.

Reviewer #1: **Yes: **Borji Rihab

Reviewer #2: No

---

## [Author Response · Author response to Decision Letter 0]

28 Aug 2020

First of all, we would like to express our gratitude to the reviewers who have taken the time to review our manuscript for possible publication in PLOS ONE. We have carefully reviewed your comments and responded to them with the aim of improving our manuscript. To aid the reviewers, we prepared a matrix to organize our responses to the comments given by the reviewers. The full matrix can be perused in the attached response letter to the reviewers.

Both reviewers have provided great feedback to further sharpen our manuscript. In particular, we wanted to highlight in this response box some of those feedback. First, the Reviewer 1 (and the editor) has raised an excellent point about determining our ID-participants' ability to provide informed consent. We have discussed our decision not to require legal representation from our ID-participants with our institutional ethics review board and we have come to agree that it also important to allow our participants to exercise their personal autonomy. In the attached response letter to the reviewers (and the manuscript itself), we have detailed our strategy to ensure that our participants understand the details related to their participation of the study. Second, we have removed Table 1, which summarized the methodological differences between stabilometric studies in ID-samples. While agree with Reviewers 1 and 2 that it can be omitted, we still believe that it may be of value to readers who may be keen on reading more about the specifics of these study (even though it may not necessarily be a complete summary). Lastly, we have also updated the limitation section of our discussion in order to reflect limitations that we may have inadvertently glossed over.

---

## [Decision Letter · Decision Letter 1]

1 Oct 2020

Reliability of center of pressure excursion as a measure of postural control in bipedal stance of individuals with intellectual disability: A pilot study

PONE-D-20-19187R1

Dear Dr. Pineda,

We’re pleased to inform you that your manuscript has been judged scientifically suitable for publication and will be formally accepted for publication once it meets all outstanding technical requirements.

Kind regards,

Robert Didden

Academic Editor

PLOS ONE

Additional Editor Comments (optional):

Reviewers' comments:

Reviewer's Responses to Questions

**Comments to the Author**

1. If the authors have adequately addressed your comments raised in a previous round of review and you feel that this manuscript is now acceptable for publication, you may indicate that here to bypass the “Comments to the Author” section, enter your conflict of interest statement in the “Confidential to Editor” section, and submit your "Accept" recommendation.

Reviewer #1: All comments have been addressed

2. Is the manuscript technically sound, and do the data support the conclusions?

Reviewer #1: Yes

3. Has the statistical analysis been performed appropriately and rigorously? 

Reviewer #1: Yes

4. Have the authors made all data underlying the findings in their manuscript fully available?

Reviewer #1: Yes

5. Is the manuscript presented in an intelligible fashion and written in standard English?

Reviewer #1: No

6. Review Comments to the Author

Reviewer #1: The paper was significantly improved. The authors have adequately addressed all comments. I recommed publication.

7. PLOS authors have the option to publish the peer review history of their article (what does this mean?). If published, this will include your full peer review and any attached files.

Reviewer #1: **Yes: **Borji Rihab

---

## [Editor Report · Acceptance letter]

12 Oct 2020

PONE-D-20-19187R1 

Reliability of center of pressure excursion as a measure of postural control in bipedal stance of individuals with intellectual disability:A pilot study 

Dear Dr. Pineda:

I'm pleased to inform you that your manuscript has been deemed suitable for publication in PLOS ONE. Congratulations! Your manuscript is now with our production department. 

Kind regards, 

on behalf of

Professor Robert Didden 

Academic Editor

PLOS ONE